# Can Eating Make Us More Creative? A Multisensory Perspective

**DOI:** 10.3390/foods10020469

**Published:** 2021-02-20

**Authors:** Qian Janice Wang, Francisco Barbosa Escobar, Signe Lund Mathiesen, Patricia Alves Da Mota

**Affiliations:** 1Department of Food Science, Faculty of Science and Technology, Aarhus University, 8200 Aarhus C, Denmark; francisco.barbosa@food.au.dk (F.B.E.); signelma@food.au.dk (S.L.M.); patricia.alvesdamota@food.au.dk (P.A.D.M.); 2Center for Music in the Brain, Department of Clinical Medicine, Aarhus University, 8000 Aarhus C, Denmark

**Keywords:** creativity, multisensory, eating experiences, cognition

## Abstract

While it is well known how food can make us physically healthy, it remains unclear how the multisensory experience of eating might influence complex cognitive abilities such as creativity. A growing body of literature has demonstrated that all human senses are capable of sparking creativity. It follows then that eating, as one of the most multisensory of all human behaviors, should be a playground for creative thinking. The present review presents an overview of how creativity is defined and measured and what we currently know about creativity as influenced by the senses, both singular and in conjunction. Based on this foundation, we provide an outlook on potential ways in which what we eat, where we eat, and how we eat might positively support creative thinking, with applications in the workplace and home. We present the view that, by offering a rich multisensory experience, eating nourishes not only our bodies but also our mental well-being.

## 1. Introduction

The need to acquire food has been suggested as a major contributor in the evolution of cognitive functions, including creativity and decision-making in humans [1]. Preliminary evidence has shown that what we eat may influence different cognitive processes [2]. For example, obesity and poor diet can lead to negative health implications including cognitive and emotional dysfunctions [2]. Furthermore, dietary problems can disrupt cognitive functions long-term. Specifically, maternal gestational diabetes may cause changes in a baby’s food reward processing [3,4,5,6]. On the other hand, a diet rich in omega-3 seems to improve cognition [7], and high consumption of fruits and vegetables has long-term benefits on cognitive function in adults [8,9]. However, the majority of research so far on diet and cognition has focused on learning and memory [2]. Given the essential role of food in the determination of brain organization [10], it is worth speculating whether the process of eating, as the ultimate multisensory activity [11,12], might also influence higher-order cognitive functions such as creativity.

### 1.1. What Is Creativity?

Creativity is a commonly used term in a broad spectrum of disciplines ranging from the arts to engineering, but it is not entirely clear what exactly people mean when they refer to creativity. Ellis Paul Torrance, known as “the father of modern creativity”, wrote:
Creativity defies precise definition. This conclusion does not bother me at all. In fact, I am quite happy with it. Creativity is almost infinite. It involves every sense—sight, smell, hearing, feeling, taste, and even perhaps the extrasensory. Much of it is unseen, nonverbal, and unconscious. Therefore, even if we had a precise conception of creativity, I am certain we would have difficulty putting it into words.(Torrance 1988, p. 43 [13])

Generally speaking, creativity is the dynamic and complex interactive process of connecting, exploring, and transforming the world in both new and meaningful ways [14]. It results from “unfamiliar combinations of familiar ideas” [15] to generate something that is novel, unpredictable, unusual, and meaningful in a specific context [15,16,17]. It is also a multimodal process that results from the integration of cues from different senses: visual, auditory, olfactory, gustatory, and tactile [18].

### 1.2. Neurological Basis of Creativity

Creative thinking is one of the most important cognitive skills [19] that allows individuals to be flexible and capable of adapting to challenges and opportunities offered by our dynamic environment [20]. Despite the fact that researchers use a heterogeneity of methodological approaches to test creativity [21], mental operations such as insight [22], conceptual expansion, overcoming knowledge constraints, creative imagery, analogical reasoning, and metaphor processing are consistently reported to be relevant in creativity [23]. Neuroscientists have used different neuroimaging methods, such as functional magnetic resonance imaging (fMRI), electroencephalography (EEG), magnetoencephalography (MEG), and positron emission tomography (PET) to uncover the neural mechanisms underlying creativity [23,24,25,26,27].

Several studies using fMRI have found creativity to be the result of a dynamic interplay between different brain networks [28,29], namely the default mode network (DMN) and the executive control network (ECN) [21,23,25,30]. The DMN has been linked to spontaneous and self-generated cognition, such as daydreaming or episodic memory retrieval [16,31]. In creativity, the DMN has been suggested to reflect the spontaneous generation of ideas acquired with the aid of long-term memory [24]. The ECN has been found to be involved in goal-directed cognition and cognitive control processes such as working memory and response inhibition in creativity studies [32,33].

### 1.3. Measuring Creativity

Creativity, as described previously, is broadly defined, and the best way to measure it still presents a challenge in research studies. However, psychometric tests have been used to test creative thinking [34]. These tests measure either convergent thinking or divergent thinking [35]. In a standard convergent thinking task, problem-solving strategies are used to reach a single solution to the problem [36]. On the other hand, in standard divergent thinking tasks, there are multiple potential solutions to a problem requiring problem-solving fluency (number of responses), flexibility (range of responses per individual), and originality (novelty of responses) in idea generation [36]. Divergent thinking (DT) involves “the retrieval of existing knowledge from memory and the combination of various aspects of existing knowledge into novel ideas” [37]. Some authors suggest the importance of both strategies in creative thinking [34], but divergent-thinking tasks remain the most commonly used for measuring creative thought [38]. For example, a recent review found that 51.1% of the neuroscience studies used divergent thinking (DT); 19.1% of the studies used convergent thinking (CT); and 29.8% tested creative performances such as drawing, writing, or musical improvisation [21]. Furthermore, researchers have shown that divergent-thinking tasks can predict expert ratings of creative performance [39] and creative accomplishments in longitudinal research [40].

To sum up, creativity is a complex cognitive process that involves a dynamic interplay between different sensory modalities and brain networks, which can be measured via neuroimaging techniques and standardized psychometric tests involving targeted problem solving as well as idea generation. However, how sensory cues from our daily experiences and environments can be harnessed to enhance the creative process is still underexplored [41,42]. To address this gap in knowledge, the following section presents an overview of research detailing how creativity is influenced by sensory cues both individually and, where applicable, in combination. Building on this overview, we develop future perspectives on how the (multi)sensory aspects involved in what we eat, where we eat, and how we eat could positively enhance creative thinking.

## 2. Sensory Influences on Creativity

We start by presenting an overview of background research on how stimuli from single or multiple sensory modalities can influence creative thinking (Table 1).

### 2.1. Visual Influences on Creativity

A number of studies have shown that vision and visual perception (physical components of the environment and how these are perceived) can significantly influence creative thinking in healthy populations. For instance, a relationship between illuminance and creativity has been reported by Wang et al. [47], who in a series of four studies examined the impact of ambient lighting on creativity. Illuminance was measured in lux (lx), which is the luminous flux of one lumen per square meter. The authors placed participants in either a dim (150 lx) or bright (1500 lx) room and used an aggregate of tasks to test both convergent (remote associates test) and divergent (originality and appropriateness of problem-solving ideas) thinking. They observed that people in the dimmer condition created more novel (but less appropriate) ideas than people in the brighter condition, ascribing this effect to the reduced inhibition resulting from being in a darker room [47]. In a similar vein, Steidle and Werth [48] investigated the effects of brightness and darkness on the creative problem-solving capabilities of their participants and found that dimmer lighting improved creative performance. Specifically, in one experiment, 114 participants were tasked with four creative insight problems and subsequently reported how free from constraints they felt. The lux/illumination level (150, 500, and 1500) varied across three sessions. The authors found that darkness increased freedom from constraints and elicited a feeling of freedom, self-determination, and reduced inhibition, which improved innovative thinking and creative performance [48].

In addition to the abovementioned research on illuminance, several studies have looked at other visual stimuli and their impact on cognition and behavior. In terms of color, evidence seems to suggest that color in the surrounding environment (e.g., in an office space) as well as color presented on computer screens influence task performance, mood, motivation, etc. [46,70,71]. In a series of studies by Mehta and Zhu [46], participants’ performance was evaluated on detail-oriented and creative tasks conducted on computers on which the background screen color was manipulated. The researchers found that warm colors (i.e., red) enhanced performance on detail-oriented tasks whereas cool colors (i.e., blue) enhanced performance on creative tasks [46]. The authors argued that this was due to the type of motivation activated by color. Specifically, red (vs. blue) is often associated with warning or danger and thus can activate avoidance motivation alongside attention to detail, accounting for the enhanced performance on the detail-oriented memory recall task. Conversely, blue is typically associated with openness, peace, and tranquillity and is more likely to activate approach behaviors, to encourage innovative, and to promote more risky strategies to problem-solving [46].

Even the way physical components in our environments are arranged and perceived can alter the way people process information. For instance, in a study by Meyers-Levy and Zhu [44], ceiling height was found to affect freedom-related (vs. confinement-related) thinking. Specifically, when the room ceiling was perceived as higher, it prompted concepts of freedom versus confinement. Somewhat contradictorily, when looking at the degree of restriction in a physical space, Levav and Zhu [72] found that individuals in a relatively confined physical space (i.e., a narrow aisle) were more likely to make more varied snack choices than people in a wide aisle. While neither study tested creativity per se, they demonstrated that participants exhibited more relational rather than item-specific thinking, when the ceiling was perceived as high (vs. low) which could enhance creativity. In addition, spatial constraints seemed to evoke reactance against an incursion to peoples’ personal space, prompting individuals to seek more variety and uniqueness in their choices [72], which could also be indicative of more creative behavior. Researchers have shown that exposure to nature, or even to natural materials, increases creative thinking [43]. The reason was that restorative environments may foster creativity and that nature has a cognitive restorative capacity.

Lastly, in an often-cited study by Fitzsimons et al. [45], it was investigated whether brands associated with creativity could enhance creative performance. In a pretest on consumer perceptions of two computer brands, Apple was believed to be more creative than IBM but was neither liked more nor perceived more positively than IBM. Following this pilot study, 341 participants were subliminally exposed to images of either the Apple or IBM logos during a visual acuity task, after which they had to complete the unusual uses task. People exposed to the Apple logo generated a significantly higher number of unusual uses, and their uses were evaluated as more creative than those exposed to the IBM logo. The researchers argued that it was the associated goals of the brands (e.g., being creative) that became activated and subsequently shaped behavior [45].

### 2.2. Auditory Influences on Creativity

Despite the fact that researchers from as early as the 1970s showed interest in understanding how background sound and music may affect creativity, there is still much to explore. Conflicting evidence indicates that environmental background sound (e.g., noise or music) may both distract creative people, leading to decreased creative performance, and increase attention, helping people gain better integration of ideas out of their focus attention and thus leading to creative thinking [73].

Early efforts investigating the relationship between creative performance and background noise includes the work of Martindale and Greenough [49], who found that a high level (75 dB) of white noise decreased performance on a convergent thinking task, namely the Remote Associates Test (RAT). The authors suggested that the higher level of noise induced higher arousal, which in turn may have been responsible for the reduced creativity. Likewise, Kasof [51] explored how background noise impacts creative thinking. He investigated the effect of loudness when background noise was predictable (vs. unpredictable), and intelligible (vs. unintelligible). The author found that loud noise negatively impacted creativity. However, an unpredictable noise had an even higher negative impact on creativity when compared to predictable noise. More recently, Mehta and colleagues [53] examined the extent to which different levels (low: 50 dB; moderate: 70 dB; and high: 85 dB) of everyday background noise (such as combined multi-talker noise in a cafeteria, roadside traffic, and distant construction noise) affected creativity performance on the Remote Associates Test. The authors found that moderate levels of everyday noise enhanced creativity when compared to low and high levels of the same noise.

Beyond noise, exposure to background music has also been linked to creative performance. For example, in a study on how music-induced mood affected creative performance, participants were asked to listen to music prior to performing a set of creative tasks [50]. The music induced different moods, such as “depressed”, “elated”, and “neutral”, and the authors found that creativity was significantly greater in participants who listened to either depressed and elated music compared to the neutral music. In a more recent study, Ilie and Thompson [52] presented participants with music of varying rate (fast/slow), pitch (high/low), and intensity (loud/soft). The researchers observed that 7-min exposure to music resulted in changes in mood, arousal, and performance on cognitive tasks. Specifically, participants who listened to high-pitch music were more successful at solving creativity tasks than participants who listened to low-pitch music. Moreover, they found that this effect was mediated by emotional valence [52].

Furthermore, two studies explored the effect of music listening while performing creativity tasks themselves [54,55]. Ritter and Fergusson [54] asked participants to listen to classical music (varying in arousal) while performing one divergent thinking task, (the Alternative Uses Task (AUT)) and three different convergent thinking tasks (the Remote Associates Test (RAT), the Idea Selection Task (IST), and the Creative Insight Task (CIT)). The authors found that participants who listened to “happy music” (classical music high on arousal and positive mood) had an increased performance in the divergent thinking task compared to participants who carried out the task in silence. However, the authors did not find the effects of music listening in convergent thinking tasks. Finally, Threadgold and colleagues [55] investigated how familiar and unfamiliar music lyrics as well as instrumental music impacted creativity performance in a convergent thinking task (one version of the RAT). The authors found that convergent thinking responses were higher in the silent condition compared to listening to unfamiliar music and impaired when listening to pop music with familiar lyrics [55].

### 2.3. Tactile Influences on Creativity

Research on the impact of haptics on creative thinking is still scarce. However, some studies have explored the influence of material hardness in tactile stimulation on creativity. For instance, in a study with 45 Chinese-speaking participants, Xie et al. [57] studied how sitting on a stool with different degrees of hardness can affect creative thinking skills. Participants sat in either a hard-surface or a cushioned stool, and they were tasked with solving a series of Chinese riddles requiring flexible/creative thinking skills as well as a series of analogical reasoning tests. The authors found that those participants who sat in the cushioned stool performed significantly better than those sitting in the hard-surface stool in the Chinese riddles test but not on the analogical reasoning tests. The authors suggested that the bodily stimulations related to material softness triggered metaphorical associations between softness and flexibility, which in turn enhanced flexible thinking. Similarly, Kim [56] found that squeezing a soft (vs. a hard) ball increased divergent thinking in 50 participants, as measured by the Torrance Tests of Creative Thinking (TTCT). On the other hand, squeezing a hard (vs. a soft ball) increased convergent thinking in 32 participants, as measured by the Remote Associates Test (RAT). The authors suggested a link between bodily experiences and types of creativity.

### 2.4. Olfactory Influences on Creativity

There is a limited number of studies exploring how smell influences creativity. For instance, Knasko [58] explored how the pleasantness of ambient odors affected creative task performance. The author asked the participants to perform the Torrance Test of Creative Thinking (TTCT) in two different sessions where the experimental room was scented either with a pleasant (lemon or lavender) or an unpleasant odor (dimethyl sulfide). The author did not find any differences in creative performance between scented or unscented sessions. However, the author noted that, when people were exposed to a pleasant odor, creative problem solving was better than when exposed to an unpleasant odor, which the authors suggested to be linked to the possibility of the odors (pleasant or unpleasant) inducing positive or negative moods. Baron and Bronfen [59] also found that pleasant odors enhanced performance on cognitive tasks requiring decoding written messages.

### 2.5. Gustatory Influences on Creativity

In terms of food and creativity, the image of drunken poets and artists is perhaps the first thing that comes to mind [74]. Experimental studies have started to investigate the effect of alcohol on creative performance. For example, one study used a vodka cranberry drink to investigate the effects of moderate alcohol intoxication on a creative problem solving task (convergent thinking). The authors found that participants who drank alcohol (blood alcohol content equal to 0.075) showed an improvement in RAT accuracy and also solved the RAT items quicker than a sober control group [61]. Benedek and colleagues used beer with and without different percentages of alcohol (0, 0.03, and 0.06) to investigate how drinking alcohol influence different measures of creativity (convergent thinking and divergent thinking) [63,75]. The authors found that lower alcohol content (BAC = 0.03) facilitated convergent thinking (RAT) performance but did not affect divergent thinking (AUT) when compared to beer without alcohol content [63]. In a later study, when increasing the number of participants, the author did not find any effects of drinking alcohol on creativity [75].

Beyond depressants, caffeinated beverages have been suggested to improve creativity based on the link between caffeine and cognition [76]. Zabelina and Silvia [66] investigated the effect of moderate caffeine consumption on different measures of creativity (convergent and divergent thinking). The authors found that participants who took a 200 mg caffeine capsule had higher performance in a convergent thinking task (The Compound Remote Associates (CRA)), but no effect in a divergent thinking test was found (The Abbreviated Torrance Test for Adults (ATTA)). Similarly, Einöther and colleagues [62] investigated the immediate effect of tea consumption on creativity and found that tea preparation and consumption improved convergent thinking (RAT) but not divergent thinking (alien drawing test). Huang and colleagues [65] found that drinking tea improved performance in two different divergent thinking creativity tests (spatial and semantic cognition).

However, it should be noted that all the above studies did not look at the gustatory/sensory aspects of the food/drinks being tested but rather revealed the influence of consuming specific chemical compounds. The question remains whether creativity can be altered by the taste of the food itself. In the only study to date documenting the influence of taste on creativity, Huh and colleagues [64] investigated how different basic tastes influence creativity. The authors found that participants listed a higher number of ideas when drinking a sweet beverage, but it was the sour taste that enhanced creative performance (i.e., participants generated more creative ideas). While the higher number of ideas could be attributed to the energizing effect of sugar, the authors did not offer an explanation for the influence of sourness on creative performance.

Finally, our emotional response to the food being eaten can also impact cognition. Isen et al. [60] demonstrated that participants who tasted a familiar branded iced tea performed better on a convergent thinking task (RAT) than participants who tasted an unfamiliar unbranded iced tea. The authors suggested that participants were transferring positive associations linked to the brand, as it was a trusted familiar brand in the geographic area of the study.

### 2.6. Multisensory Influences on Creativity

In terms of creativity enhancement, are multiple senses better than one? In the area of creativity support tools, Goncalves and Campos [68] demonstrated that a creative writing support software with both audio (inspirational soundtrack) and visual (dynamic landscape) stimuli promoted greater self-reported creativity compared to using a baseline text processor without any audiovisual accompaniments. Interestingly, the authors found that a “just enough” approach providing audiovisual stimuli within the software itself was at least as good, if not better, at promoting creativity than a full-on virtual reality environment. However, it should be kept in mind that the authors did not study the auditory or visual stimuli in isolation. Therefore, it is difficult to conclude whether the audiovisual combination would have worked better than either sense by itself. That said, there is evidence suggesting that fewer might be better when it comes to stimulating creativity. In a study with elementary school children, Greenfield et al. [67] found that children who listened to a partial story in radio format were able to come up with more imaginative story completions compared to when they listened to the story in television format. The researchers hypothesized that radio, by lacking visual content, stimulated the children’s visual imagery, whereas television inhibited it (see [77] for a review on how television reduces creative imagination).

In the only example of a multisensory study involving chemical senses, the combination of relaxing and stimulating smells and music was introduced to high school students during a writing task [69]. Interestingly, the combination of relaxing smell (laurel) and music (nature sounds) led to the highest creativity support index scores compared to either sense presented alone. However, the combination of a stimulating smell (coffee) and sounds (noisy cafe) had in fact a negative effect on the impression of creative support due to the fact that the combination of alerting smell and sound were reported to be overwhelming.

In summary, we have cause to believe that the right type of multisensory intervention might boost creativity if we are mindful of the type of creativity we want to promote and avoid overloading the senses. Perhaps the combination of sensory stimuli that naturally go well with each other (e.g., audiovisual stimuli from a creative writing software) may pair better than the relatively more unusual combination of sound and smell. Given the broad spectrum of evidence regarding how different types of sense input support creativity, we now outline ways in which the multisensory rich environment of eating might promote flexible thinking. We break down the process of eating to focus on how intrinsic food properties, the extrinsic eating environment, and the mindset of the eater can all separately promote creativity. Finally, we present an overview of the implications of promoting creativity for the workplace and home.

## 3. Future Perspectives: How Might the Sensory Experience of Eating Influence Creativity?

Given that eating is one of the most multisensory experiences in life [11], it is worth speculating how multisensory interactions inherent in the eating experience might influence creativity. Based on the literature review above, the senses can stimulate creativity in a variety of ways. First, sensory stimuli that are widely associated with creativity, such as the Apple logo, can promote divergent thinking [45]. In a related area, even sensory stimuli that prime concepts related to creativity, such as freedom [44] or flexibility [57], has shown similar enhancement effects. Moreover, creativity appears to depend on one being in a relaxed and positive emotional state. Therefore, sensory stimuli that promote a positive mental state also enhance creativity, whether it is from relaxing colors [46], happy music [54], familiar foods [60], or the combination of relaxing odors and music [69].

Beyond these principles, we can also refer to the multimodal nature of creativity, which results from the integration of different sensory cues present in the environment arising from the different human senses: visual, auditory, olfactory, gustatory, and tactile [18]. Creativity involves making meaning from different elements in new ways, and as we will demonstrate later, this thought process can be applied to the dining table as well.

When applying creativity principles to the dining table, it is worth remembering that the process of eating goes beyond the food itself [12]. In the sections below, we address, in turn, how factors related to the food itself (food-intrinsic), to the eating environment (food-extrinsic), and to the mind of the eater (psychological) might be modified to promote creativity (Table 2). Focusing on the multisensory perspective of the review, this section will be limited to discussions related to the sensory aspects of the holistic eating experience. The physiological and social aspects of eating are well-researched areas with demonstrated influences on cognition and mood (see [78,79,80,81], for reviews). While these aspects can potentially influence creativity and should be considered in future studies relating to creativity and eating, they are out of the scope of the current review.

### 3.1. Intrinsic Food Properties

Starting with the sensory aspects of the food itself, we examine the different ways in which what we eat might impact creativity.

#### 3.1.1. Complexity

If creativity is based on the process of discovering heretofore unknown links between different areas/sensory attributes [82], then consuming foods that offer a more diverse sensorial experiences should give more opportunities for inspiration. Certainly, environments high in visual complexity are considered to demonstrate higher creative potential [43]. Many foods, such as coffee, tea, and wine, are known to possess potentially high levels of complexity [83]. While complexity in the chemical senses can be defined in a variety of ways, one commonly accepted interpretation refers to the number of flavors perceived by the taster [84]. Therefore, one wonders whether the myriad of studies [60,62,65] showing improved cognitive performance from drinking tea can be explained by caffeine alone or whether the sensory aspects of consuming a complex beverage additionally boosts creative thinking. Further studies could investigate the perceived complexity as a variable. For instance, could more complex single-origin specialty coffee confer more creative benefits than the standard carafe of discount coffee so often found in workplaces?

Given that the way people judge complexity differs by expertise level [85,86], one question is whether food complexity needs to be explicitly recognized. Alternatively, might it be whether it is enough to just passively perceive the variety of sensory stimuli in a food. Maybe a certain level of expertise with the product is required before one can fully gain the benefit from tasting a complex food.

#### 3.1.2. Emotion

As our literature review in Section 2 revealed, many examples of sensory influences on creativity operated under the mechanism of emotion mediation. Creativity is enhanced when people feel happy [54,58,59] and either relaxed [69] or moderately excited [52,53]. Considering the mounting research on how what we eat influences how we feel (see [78,87], for reviews), future studies should investigate if foods with sensory characteristics that make us feel relaxed and happy, such as sweet foods [88] or soft foods [89], might also help us become more creative?

Another avenue for future research relates to food familiarity, although there seems to be contradictory evidence regarding the effect of stimuli familiarity on cognitive performance. While at first glance, familiarity, with its common-place associations with routine and boredom, would seem to be detrimental to creativity, there is evidence that consuming a familiar branded iced tea improved performance on a convergent thinking task [60]. In this case, perhaps the more familiar beverage helped participants feel more relaxed in an otherwise stressful laboratory setting. On the other hand, it is easy to imagine that, in a real-world setting, workplace or school canteens may want to serve novel foods once in a while to avoid employees and students falling into a routine. Taken together, it is clear that more research is needed in this area in order to uncover the conditions in which familiarity might help creative thinking.

#### 3.1.3. Conceptual Priming

Finally, given that many chefs now pride themselves on serving creative dishes [90], one might ask then if merely eating creative dishes might inspire the diner to be more creative? Similar to exposure to the Apple logo [45], eating a dish from a restaurant known for its creativity might implicitly prime the diner. This would especially be the case if one believes in the power of ingestion to transfer metaphorical powers [91]. Moreover, beyond specifically priming creativity, we can also consider how foods could express related concepts such as freedom or flexibility. These concepts might be related to where the food comes from (e.g., free-range eggs) to the properties of the food itself (e.g., flexible foods such as jello). Just as sitting on a soft surface can improve divergent thinking [57], might eating soft foods do the same?

### 3.2. Food-Extrinsic Factors

We constantly receive an influx of sensory signals that arise from many different sources in the environment. The nature of our surroundings plays an important role not only in how we feel but also on how we perform tasks [92], be it for work or leisure or, as in this case, for eating. Research has shown that the eating environment can influence what the food tastes like and how much we like it [12]. In other words, extrinsic contextual factors play an important role in the eating experience. In the context of creativity, it is first worth considering how a well-designed dining environment might passively encourage creativity just from the incidental combination of sensory stimuli present in the environment. If so, eating in an environment that combines creativity-promoting attributes (Table 1) should induce creativity. For instance, eating in a blue-colored dining room with soft seats and happy music should make one more creative compared to, say, a red-colored room with hard seats and loud noise. Work by McCoy and Evans [43], who asked participants to evaluate the creative potential of visual spaces, revealed that environments with high creativity potential featured exposure to nature or natural elements, spatial complexity, visual detail, and opportunities for social interaction. In general, when designing a creative eating environment, one should aim to induce a positive, nurturing, and social mindset and/or to introduce design elements with a clear association to creativity or related elements (e.g., tall ceilings to convey freedom or dim lighting to reduce inhibition). One potential area of future research involves investigating how multisensory combinations of creativity boosting factors can be combined in the eating environment while avoiding giving diners sensory overload [69].

### 3.3. Mindful Eating

The way people actively think about/evaluate the food can influence creativity. If creativity is about drawing information from the different senses, then a methodological examination of the different senses should already induce creativity.

People do not usually pay attention to the food they eat [93,94]. However, in specialized fields such as wine and coffee, professional tasting procedures involve focusing one’s attention to the different sensory modalities: vision, smell, taste, and mouthfeel. For example, the Wine and Spirits Education Trust, one of the largest global providers of wine education, uses a guideline called the Systematic Approach to Tasting Wine (https://www.wsetglobal.com/knowledge-centre/wset-systematic-approach-to-tasting-sat, accessed on 15 January 2021) which divides wine evaluation to appearance, nose, and palate—with evaluation criteria for both taste (e.g., sweetness and acidity) and oral-somatosensation (e.g., body and tannin). Similarly, the Specialty Coffee Association cupping protocol (https://sca.coffee/research/protocols-best-practices, accessed on 15 January 2021) involves separate assessments for fragrance, taste (e.g., acidity and sweetness), and mouthfeel (e.g., body).

Furthermore, metaphors are common in descriptions in complex foods such as wines [95,96], where figurative language is often brought in to help people communicate aromas and flavors. For instance, tasting notes often use anthropomorphic language when describing a wine, producing examples such “a monster in a beautiful frock… loads of velvety tannins” [96]. Beyond producing tantalizing descriptions, using metaphors to describe foods may in fact enhance creative thinking as well. Evidence shows that reading a narrative poem containing open metaphors boosted divergent thinking compared to reading an appliance manual [97], so there is reason to believe that actively generating metaphorical associations based on the tasting experience may do the same.

We recently put this theory to test at a specialty coffee sensory conference, where 225 coffee professionals answered an online survey. All the participants were sent a brew-your-self coldbrew kit and prepared the same coffee before activating the online survey. All the participants first answered a convergent thinking (RAT) and divergent thinking task (AUT) and, then, tasted and evaluated a coffee sample before answering another set of convergent and divergent thinking tasks. Half the participants evaluated the coffee based on the Specialty Coffee Association’s cupping protocol (https://sca.coffee/research/protocols-best-practices, accessed on 15 January 2021), while the other half evaluated the coffee with the cupping guide as well as rated how the coffee matched nine pairs of metaphors on a seven-point semantic differential scale (Figure 1). We hypothesized that the activation of all the senses in the novel metaphor condition would further boost creativity, compared to those who only rated the coffee according to the cupping guide. The results demonstrated a significant improvement in convergent thinking before and after tasting in the metaphor group (*F*(1, 113) = 10.43, *p* = 0.002), while no significant group-based differences were found in divergent thinking (*F*(1, 95) = 0.05, *p* = 0.827). In other words, the process of coming up with multisensory metaphors for the coffee one happened to drink improved convergent thinking.

## 4. Implications of Creativity for the Workplace and Home

The findings of the present review reveal the influence of multisensory integration on creativity. A relevant practical application of these findings lies in the creation of spaces and activities that leverage multisensory integration to promote organic creative thinking in the personal and professional realms. Given the technological advancements and increasing flexibility of workspace locations, the creation of these spaces is relevant for offices, social places, and homes.

Creativity in organizations is an important focus of research in organization sciences [98]. Previous literature has found a positive relationship between employee creativity and firm growth rates [99]. This line of research demonstrates the relevance for firms to invest in the promotion of divergent thinking. However, organizational creativity should be managed carefully since poorly designed strategies can be counterproductive and can hinder creativity itself [100]. Hence, the development of activities and spaces that exploit multisensory integration is a potential organic strategy that can foster a creative environment and can improve firm performance.

Firms can adopt strategies with different levels of investment and engagement to exploit multisensory interactions to foster creative thinking. As a passive approach, firms can change specific elements of physical spaces. For instance, firms can change the color schemes of offices, common areas, and meeting rooms to incorporate more blue tonalities. Moreover, the material of chairs across the workplace can be changed with softer textures. Sonic environments with happy music can be implemented in select places. A more active approach may leverage complex food and tastings in different scenarios, from food at canteens to dedicated tasting events. Strategies such as these would allow employees to actively engage with the multisensory elements of the food and the environment and to socialize with their colleagues.

The present findings are also relevant for coffee shops and co-working spaces. Since the mid-sixteenth-century, coffee shops have provided a social space for creative discourse and activities [101]. Nowadays, coffee shops continue to play a myriad of roles for businesses and communities, including creative organized activities as well as working and networking spaces [102]. The share of people working and studying remotely in these types of spaces has grown rapidly [103,104]. Therefore, these businesses have a unique opportunity to leverage the influence of multisensory integration on creativity to increase visits, engagement, and profits. Coffee shops may develop special environments that stimulate multiple senses and make people mindful of these inputs and their interactions. For instance, coffee shops can manipulate environmental colors and lighting as well as the sonic background and the textures of the seats and tables. Besides the environment, coffee shops can offer complex beverages and can nudge consumers to be mindful of their flavor and textural complexity. Potential inspiration for the latter strategy can be drawn from many specialty coffee roasters that present their coffee with stories about the terroir and the farmers, tasting notes, and often metaphorical associations. Co-working spaces can adopt blended strategies for regular office spaces and coffee shops and can add complex foods and beverages to their list of amenities.

Finally, given the increasing trend of people working from home, in addition to the immediate and potential future effects of the COVID-19 on work culture, our findings have especially relevant applications for remote work. People may change physical aspects of their home related to the color and materials of their working spaces. Additionally, sonic environments can be easily created and modified according to specific situations. Furthermore, people may dedicate time to practice mindful tastings of different foods.

## 5. Conclusions

In this paper, we reviewed recent developments in sensory-based creativity research. Research has shown that sensory information from all sensory modalities can boost creativity in different ways: emotion modulation (e.g., [58]), priming with creativity (e.g., [45]) or related concepts (e.g., [57]), and helping people find unexpected connections ([97]). Taken together, it is clear that multisensory experiences have the potential to change the way we think.

There has been a dearth of research on the creativity boosting potential of eating, one of the most multisensory daily activities. Guided by previous research, we made recommendations/suggestions for ways in which what we eat, where we eat, and how we eat can induce greater creativity (Table 2). Moreover, the potential to increase creativity via our eating experiences has major implications for the workplace and home. Therefore, we conclude that this is an exciting area of interdisciplinary research that deserves future attention.

## Figures and Tables

**Figure 1 foods-10-00469-f001:**
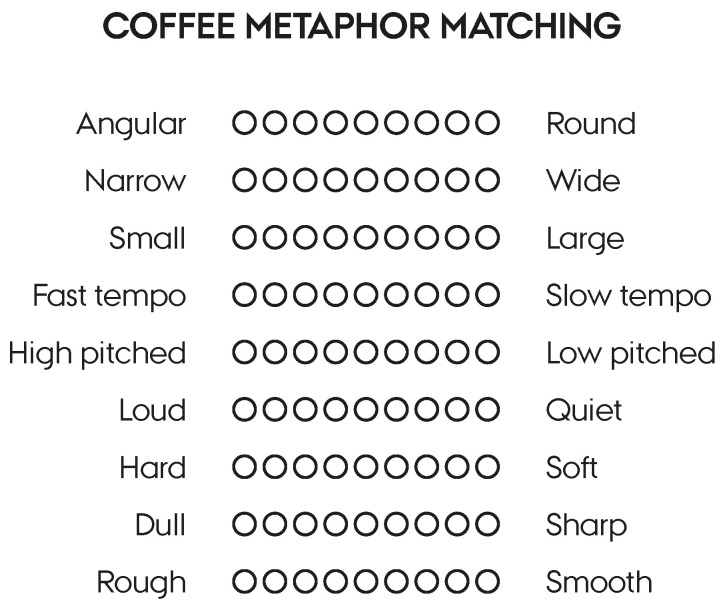
Metaphor pairs used in the coffee creativity study at the Sensory Summit US (2020).

**Table 1 foods-10-00469-t001:** Published research on uni- and multisensory influences on creative thinking.

Study	Modality	Stimuli	Findings	Mechanism
McCoy and Evans (2002) [43]	Vision	Environments with different features	Environments with natural views and use of natural materials have higher perceived creativity potential	Exposure to nature restores cognitive capacity
Meyers-Levy and Zhu (2007) [44]	Vision	High vs. low ceiling height	Higher ceiling promotes relational thinking	Higher ceiling primes concept of freedom
Fitzsimons et al. (2008) [45]	Vision	Exposure to brand images	Creative brands promotes creativity	Associated brand goals activate behavior
Mehta and Zhu (2009) [46]	Vision	Computer screen color	Blue enhances creativity (red enhances memory recall)	Color activates approach/avoidance (blue is associated with approach, and red is associated with avoidance)
Wang et al. (2011) [47]	Vision	Dim vs. bright illuminance	Dim light enhances creativity	Dim light reduces inhibition
Steidle and Werth (2013) [48]	Vision	Dim vs. bright illuminance	Dim light enhances creativity	Dim light promotes freedom from constraints
Martindale and Greenough (1973) [49]	Audition	Noise inducing low (relaxed), medium (stress), and high (white noise) arousal	High arousal (75 dB white noise) impacts creative performance	Lower levels of arousal facilitates creative performance
Adaman and Blaney (1995) [50]	Audition	Music inducing “elated”, “depressed”, or neutral moods	Greater creativity after listening to depressed and elated music	Mood change is associated with higher creativity
Kasof (1997) [51]	Audition	Noise vs. quiet place	Noise (intelligible or unpredictable) impaired creative performance	Exposure to arousal stimuli reduces breadth of attention.
Ilie and Thompson (2011) [52]	Audition	Same musical piece varied in intensity, rate, and pitch height	Greater creativity at high pitches than in low pitches	Effect of pitch height mediated by emotional valence
Mehta et al. (2012) [53]	Audition	Background noise with low (50 dB), moderate (70 dB), and high (85 dB) levels	Moderate level (vs. low) of noise enhances creativity; high level of noise impacts creativity	Moderate and high noise levels lead to abstract processing, with the higher level reducing information processing
Ritter and Ferguson (2017) [54]	Audition	Classical music with different levels of arousal vs. silence	Happy music increased divergent thinking but not convergent thinking	Flexible thinking style helped participants come up with more creative ideas
Threadgold et al. (2019) [55]	Audition	Familiar vs. unfamiliar music with lyrics vs. instrumental vs. silence	Convergent thinking was higher in silence compared to all the other conditions	Changing states of sound in music disrupts verbal working memory processes
Kim (2015) [56]	Touch	Hard vs. soft ball	Soft material improves divergent thinking	Bodily experience of softness influences creative thinking
Xie et al. (2016) [57]	Touch	Hard-surface vs. cushioned stool	Soft textures improves creative thinking	Material softness triggers metaphorical associations with flexible thinking
Knasko (1992) [58]	Olfaction	Pleasant vs. unpleasant odors	Better creative problem solving when exposed to pleasant odor	Improvement in mood induces problem solving
Baron and Bronfen (1994) [59]	Olfaction	Pleasant fragances vs. no odor	Pleasant fragance enhanced performance on cognitive tasks involving creativity.	Pleasant fragrances induce positive affect
Isen et al. (2004) [60]	Gustation	Familiar vs. unfamiliar brand of iced tea	Better performance on convergent thinking after drinking a familiar brand of iced tea	Familiar brand name induces positive affect
Jarosz et al. (2012) [61]	Gustation	Vodka with 0.075 vs. control	Alcohol (0.075) improved convergent thinking	Inhibition and less attentional control leads to better associative approaches
Einöther et al. (2015) [62]	Gustation	Tea vs. water	Tea preparation and consumption improved convergent thinking but not divergent thinking	Positive affect leads to more associative and flexible processing style
Benedek et al. (2017) [63]	Gustation	Beer with alcohol (0.03) and placebo	Alcoholic beer (0.03) facilitated convergent thinking but did not affect divergent thinking.	Alcohol intoxication may reduce fixation effects by loosening the focus of attention
Huh et al. (2018) [64]	Gustation	Sweet vs. sour drink	Sour taste enhanced creative performance	Not stated in paper
Huang et al. (2018) [65]	Gustation	Tea vs. water	Drinking tea improved performance in two divergent thinking tests	Drinking tea increases mood valence (positive affect)
Zabelina and Silvia (2020) [66]	Gustation	Capsule of caffeine (200 mg) vs. placebo	Caffeine improved performance in convergent thinking, but no effect was found in divergent thinking	Enhanced concentration and attentional focus
Greenfield et al. (1986) [67]	Vision and audition	Television vs. radio	Children made more imaginative story completions with radio presentation compared to television	Radio stimulated visual imagery
Goncalves and Campos (2018) [68]	Vision and audition	Creative support software with both audio and visual components	Audiovisual stimuli promoted greater self-reported creativity compared to baseline text processor	Audiovisual stimuli provided “just enough” immersion in another environment
Goncalves et al. (2017) [69]	Audition and olfaction	Relaxing/stimulating aromas and sounds	Relaxing aroma plus music enhanced sense of creativity support compared to either sense alone	Sensory combination induced more relaxation

**Table 2 foods-10-00469-t002:** A summary of ways in which the sensory aspects of the holistic eating experience might enhance creativity.

Creativity Mechanism	Food Intrinsic Factors	Food Extrinsic Factors	Psychological Factors
Priming via associations with creativity (e.g., logos)	Creative cooking	Furniture/servingware by known creative designers Artworks in dining room Creative music (e.g., improvised)	
Priming via concepts related to creativity (e.g., freedom and flexibility)	Food origin (e.g., free-range eggs) Soft foods (e.g., jello)	Soft seats and table linings Tall ceilings Dim lighting	
Positive and relaxed mental state	Liked foods Sweet foods Familiar foods	Positive relaxing music, colors, and fragrances Nature/natural elements Encourage social interaction	
Exposure to unforeseen connections	Complex foods		Structured tasting that focus on all sensory modalities Describing food with metaphors

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
