# Peer review of "Can Eating Make Us More Creative? A Multisensory Perspective"

_foods, 2021, doi:10.3390/foods10020469_

Round 1
Reviewer 1 Report
Here are my detail comments:
The present manuscript submitted by Wang et al. provides an outlook on potential ways in which what we eat, where we eat, and how we eat might positively support creative thinking, with applications in the workplace and home. This is an emergent topic on the field and plenty of originality. So, the developed work is justified and this is well evidenced in the Introduction section.
The authors have written the manuscript quite well according to the journal's instructions and well structured. It is also easy to read and the conclusions are consistent with the evidence and arguments presented before.
I only have to ask the authors to clarify the main goals of the manuscript at the end of Introduction. Thus, readers must be clearly informed about the final answer to the question posed initially.
Table 1 should be posed near the line 88, where this table is mentioned in the text.
Reviewer 2 Report
The authors explore the possibility that eating can somehow facilitate creativity in the consumer. To this end, the authors review a vast literature on factors that affect creativity, such as music and colors. There are indications that multisensory experiences facilitate creativity. It is therefore logical that eating, an extreme multisensory experience that includes our tactile, visual, taste, and smell, and sometimes even auditory faculties, may be the perfect tool to boost creativity. A potential complication with eating is however that eating has psychological – just like colors and music- as well as physiological consequences. These physiological consequences – e.g. effects on satiety hormone and on gastric emptying, may have dramatic effects on one’s functioning, and especially on one’s mood. This has been extensively demonstrated by studies with for example with high fat or high carbohydrate foods, or chocolate. Surprisingly, this literature is mostly lacking in this literature review, even though effects of other forms of stimulation on mood and emotions are mentioned. I strongly recommend adding relevant literature from food science to the review.
The authors’ review of existing literature provides an overview of effects of various factors on creative measures, but also on other measures such as emotions, but possible relationships between these measures is not further explored. This excludes the possibility that possible effects on creativity are mediated by for example effects on mood and emotions. Intuitively, I would think that my creativity is low when I am in a foul mood. Hence, creativity may not be a factor such as mood or arousal, but fluctuations in creativity may actually result from fluctuations in mood. This distinction is important because it would give direction to possible studies on creativity. Also, I would like the authors to expand on the possible psychological and physiological effects during eating, and how these effects may be experimentally separated, e.g., by comparing the effect of actual eating with the effect of sham-eating (i.e., chewing without digestion. Also, the eating itself may not be important for creativity, but the social and physical circumstances in which food is consumed, may be more important.
Round 2
Reviewer 2 Report
I thank the authors for their clarifications and revisions.